# OpenReview forum: "T2VUnlearning: A Concept Erasing Method for Text-to-Video Diffusion Models"
_ICLR.cc/2026/Conference — ICLR 2026 Conference Withdrawn Submission_

### Official Review · Reviewer_n2NF · 2025-10-20

**Soundness:** 3
**Presentation:** 2
**Contribution:** 2
**Rating:** 2
**Confidence:** 4

**Summary:**

The paper proposes an approach to erase concept from T2V models. They do so using a combination of negatively guided velocity predictions, mask based location and a preservation loss on unrelated concepts. They show results on nudity, object and faces.

**Strengths:**

- The authors have adopted various T2I concept erasure techniques into T2V models and show that it works.
- They show promising results on various tasks including nudity, objects and faces.
- The paper is generally well written and I appreciate that the authors have openly cited works from which they have borrowed particular ideas.

**Weaknesses:**

- The authors seem to be making a distinction between unlearning methods for T2I vs T2V diffusion models. Most methods proposed for T2I models could directly also be applied to T2V models. Thus I think claims such as "we are the first to propose an unlearning-based concept erasing method for T2V models" need to be revisited. Especially given methods such as SAFREE have already shown generalizability to T2V models.
- The proposed method is an adaptation of ESD and Receler for T2V models.
- The baseline comparisons are lacking. While the paper proposes a weight update based method it only compares itself to inference based methods in Table 1.
- Since there are so many red-teaming efforts now to bypass concept erasure, I am particularly interested in seeing how this performs on that but this is currently missing from the paper.

**Questions:**

- How does the method perform with multi-concept erasure?
- Which Ring-A-Bell dataset have you used? Is it specific to T2V models also does it contain adversarial prompts like the original version or the chat-gpt prompts like the version from SAFREE?

---

### Official Review · Reviewer_HJ6n · 2025-10-22

**Soundness:** 2
**Presentation:** 2
**Contribution:** 1
**Rating:** 2
**Confidence:** 5

**Summary:**

This paper proposes a negatively-guided framework for concept unlearning in text-to-video (T2V) diffusion models. The method extends text-to-image (T2I) unlearning strategies to the video domain by combining negative velocity guidance, temporal masking, and LLM-based prompt augmentation. Experimental results on nudity and identity removal are presented.

While the problem of safe and controllable video generation is important, the proposed approach suffers from several fundamental issues in task formulation, methodological choice, and experimental evaluation. Overall, I find the technical novelty and empirical validation insufficient for acceptance.

**Strengths:**

- Addresses an emerging and relevant topic — safety and concept erasure in T2V generation.
- Attempts to explore an unlearning perspective beyond simple prompt filtering.
- Provides qualitative examples and limited user study.

**Weaknesses:**

1.	Unreasonable setting and problematic methodology.
The key distinction between T2V and T2I lies in temporal consistency. Video-level concept unlearning could naturally exploit temporal priors such as keyframe guidance or inversion-free conditioning. In contrast, the proposed negative-guidance approach acts purely on the noise/velocity level, which is not a principled choice for video data. This design inherits known instability and over-suppression issues from prior image-level works, leading to numerous ad-hoc “tricks” to mitigate fundamental flaws rather than solving them.

2.	Unconvincing and low-quality results.
The qualitative results are not persuasive.
- Targeted unlearning often degenerates into object removal (e.g., Fig.8 removes the entire person instead of only nudity).
- Even after multiple auxiliary tricks, the model still forgets irrelevant semantics.
- For non-targeted unlearning, results remain almost identical (especially in faces), suggesting weak control ability.

3.	Biased and insufficient user study.
The user study questions are likely leading and lack fine-grained criteria. For example, “nudity removal” should evaluate whether the person remains but becomes clothed, not whether the person disappears. The study therefore cannot reliably support the claimed effectiveness.

4.	Absence of video results.
As a T2V work, it is unacceptable that no video samples or temporal metrics are shown. The absence of video demonstrations raises strong suspicion that the model’s tricks harm temporal coherence and visual creativity. Quantitative metrics and video demos are essential.

5.	Questionable LLM-based prompt augmentation.
LLM-generated text augmentation is not necessarily aligned with the T2V text-encoder embedding space, and may bring LLM-specific fitting effects. Robustness against different LLMs or jailbreak prompts is untested. A more principled and robust approach would be embedding-space adversarial perturbation rather than text-level augmentation.

6.	Mitigation tricks leading to trivial solutions.
The added preservation regularizers and masks appear to suppress removal artifacts but actually encourage trivial “erase-the-object” solutions, harming fine-grained unlearning quality.

7.	Missing comparisons to keyframe-guided or training-free baselines.
Since the goal is unlearning without retraining, it is crucial to compare against inversion-free or keyframe-based editing approaches to justify the proposed methodological path. Without this, the claimed advantage over simpler alternatives remains unsubstantiated.

**Questions:**

1.	Can you provide video samples and temporal quality metrics?

2.	How does your method compare to keyframe-guided or training-free editing for concept removal?

3.	Have you tested LLM augmentation across different LLMs or using adversarial embedding perturbation?

4.	What measures were taken to ensure the user study questions were unbiased?

---

### Official Review · Reviewer_VAQJ · 2025-10-27

**Soundness:** 2
**Presentation:** 2
**Contribution:** 2
**Rating:** 4
**Confidence:** 4

**Summary:**

The paper proposes T2VUnlearning, an adapter-based fine-tuning approach to erase undesirable concepts from text-to-video (T2V) diffusion models while preserving non-target capabilities. The method combines:
(1) negatively-guided velocity prediction (a reparameterization of score-based negative guidance) enhanced with LLM-based prompt augmentation;
(2) mask-based localization that extracts concept masks from text–video regions in full-attention QK maps to spatially constrain updates; and
(3) concept-preservation regularization (inspired by prior-preservation in DreamBooth) to mitigate catastrophic forgetting.

**Strengths:**

Comprehensive evaluation: multiple model families, diverse prompt distributions (including human-written), and a mix of automatic and human studies. The SafeSora and VBench analyses are appropriate, and the face-erasure study is a challenging stress test.

**Weaknesses:**

1. VBench Object Class and Subject Consistency are valuable, but broader “video quality” and “text-video alignment” metrics (e.g., aesthetic/FLA, motion fidelity, temporal consistency beyond a single metric) could reveal subtle degradations post-unlearning.
2. The augmentation is crafted to mirror T2V training prompts; robustness to adversarial or diverse paraphrases outside the LLM’s style remains uncertain. Reporting performance under adversarially optimized prompts (beyond long/refined ones) would be useful.
3. The thresholding strategy for QK-based masks appears heuristic. Sensitivity to thresholds, layer selection, and temporal consistency of masks is underexplored. Failure modes when concepts are diffuse (styles, activities, or abstract attributes) are not deeply analyzed.
4. Although adapters are lightweight, the reported 300–500 epochs per target concept may still be costly for practitioners who need to erase many concepts or maintain fleets of models. Comparisons to training-free methods [1,2] in wall-clock and energy would help position the method.
5. Choosing a single “preserve” concept (e.g., “person” for nudity) is intuitive, but may not cover broader semantic neighborhoods. It is unclear how well this scales when multiple related concepts need protection simultaneously. This is also discussed in ANT [3].
6. Potential regrowth or circumvention: The paper does not study whether erased concepts re-emerge after continued fine-tuning on benign data, or whether jailbreak-style prompting can bypass the erasure boundary.
7. The number of compared baselines is too limited. It would be beneficial to include some classic methods (such as MACE [4] and AC [5]) to make the comparison more comprehensive.
8. Can the proposed method be applied to the FLUX model, similar to what EraseAnything [6] does?

[1] SAFREE: Training-Free and Adaptive Guard for Safe Text-to-Image And Video Generation

[2] Concept Corrector: Erase concepts on the fly for text-to-image diffusion models

[3] Set You Straight: Auto-Steering Denoising Trajectories to Sidestep Unwanted Concepts

[4] MACE: Mass Concept Erasure in Diffusion Models

[5] Ablating Concepts in Text-to-Image Diffusion Models

[6] Enabling Concept Erasure in Rectified Flow Transformers

**Questions:**

See weaknesses.

---

### Official Review · Reviewer_gWkN · 2025-10-31

**Soundness:** 3
**Presentation:** 2
**Contribution:** 3
**Rating:** 6
**Confidence:** 2

**Summary:**

This paper proposes T2VUnlearning for text-to-video diffusion using DiT/MMDiT backbones. This method involves inserting lightweight adapters after full-attention layers and fine-tuning them via negatively-guided velocity prediction coupled with mask-based localisation regularisation and concept-preservation regularisation. LLM-based prompt augmentation is also employed to enhance robustness against prompt reformulations. Experiments on CogVideoX-2B/5B and HunyuanVideo demonstrate the effective suppression of targeted concepts (e.g. nudity, object categories and celebrity/identity) while preserving non-target content to a large extent.

**Strengths:**

1. Problem significance. The systematic transfer of concept unlearning from T2I to T2V addresses pressing safety and compliance needs in generative video.
2. Engineering practicality: Plug-and-play adapters are applicable to multiple public T2V backbones, and the approach appears deployment-friendly.
3. Relatively comprehensive evidence. It covers three sensitive concept families, multiple models and metrics, plus ablations and a user study.
4. Clear localisation idea. It uses QK interactions with full attention to approximate text–visual alignment regions, thereby reducing unintended forgetting outside the target area.

**Weaknesses:**

1. The paper criticises the reliance on LLM-refined prompts for inference in prior SOTA for undermining defences, yet later adopts LLM-based prompt augmentation in training. Please clarify the distinction in terms of stage, objective and risk, and articulate the novel contribution beyond a combination of known components.
2. Nudity evaluation largely depends on a single detector (e.g. NudeNet), so bias/misclassification may influence conclusions. There is a lack of multidetector agreement or small-scale human calibration.
3. The mask relies on text–visual interactions requiring full attention, and its robustness to sparse/block attention, multi-scale attention or variant implementations has not been demonstrated.

**Questions:**

Have you tried multi-detector voting or small human-labeled subsets to validate the stability of the Nudity Rate? Please quantify how much the main conclusions change under different detectors/thresholds.

---

### Note · Authors · 2025-11-12

I have read and agree with the venue's withdrawal policy on behalf of myself and my co-authors.